# Effect of the Composition of Copolymers Based on Glycidyl Methacrylate and Fluoroalkyl Methacrylates on the Free Energy and Lyophilic Properties of the Modified Surface

**DOI:** 10.3390/polym14101960

**Published:** 2022-05-11

**Authors:** Viktor V. Klimov, Olga V. Kolyaganova, Evgeny V. Bryuzgin, Alexander V. Navrotsky, Ivan A. Novakov

**Affiliations:** 1Chemical Engineering Faculty, Volgograd State Technical University, Lenin Avenue, 400005 Volgograd, Russia; vicklimov@gmail.com (V.V.K.); ollik86@mail.ru (O.V.K.); navrotskiy@vstu.ru (A.V.N.); president@vstu.ru (I.A.N.); 2Department of Chemistry, Lomonosov Moscow State University, Leninskiye Gory, 1, Building 3, 119991 Moscow, Russia

**Keywords:** superhydrophobicity, glycidyl methacrylate and fluoroalkyl methacrylate copolymers, surface free energy, textured aluminum, sliding angles measurements, self-cleaning effect

## Abstract

This study proposes to use reactive copolymers based on glycidyl methacrylate and fluoroalkyl methacrylates with a low fluorine content in the monomer unit as agents to reduce the surface free energy (SFE). This work reveals the effect of the structure and composition of copolymers on the SFE and water-repellent properties of these coatings. On a smooth surface, coatings based on copolymers of glycidyl methacrylate and fluoroalkyl methacrylates with fluorine atoms in the monomer unit ranging from three to seven are characterized by SFE values in the range from 25 to 13 mN/m, which is comparable to the values for polyhedral oligomeric silsesquioxanes and perfluoroalkyl acrylates. On textured aluminum surfaces, the obtained coatings provide time-stable superhydrophobic properties with contact angles up to 170° and sliding angles up to 2°. The possibility of using copolymers based on glycidyl methacrylate and fluoroalkyl methacrylates for the creation of self-cleaning polymer coatings is shown.

## 1. Introduction

At the present stage of technological development, the world is witnessing a transition to cost-effective, resource-saving, and energy-saving technologies that save natural resources and avoid environmental pollution, e.g., the use of secondary resources, self-cleaning materials, and coatings [1,2,3]. Thus, one of the high-priority areas of research is a directed change of properties at the interface, which can be used to control the lyophilic characteristics of the surface and, in particular, to impart water-repellent properties to common materials [4,5,6]. Many structural materials and alloys operating in high humidity conditions are subject to corrosion, biofouling, and icing. Dust deposition on the surfaces of structures or special devices is associated with significant cleaning costs and in some cases reduces performance, e.g., of solar energy conversion devices and telecommunication antennas [7]. Therefore, the creation of self-cleaning coatings with stable hydrophobic properties is a high-priority area of research [8,9,10].

The fundamental principles for creating such materials are based on the works of Young and Wenzel, Cassie, and Baxter [11,12,13]. Over the past few decades, there has been a steady increase in publications dedicated to the creation of coatings with superhydrophilic or superhydrophobic properties. The contact angle (CA) is commonly used as a characteristic of wettability [14,15]. Surfaces with a contact angle less than 90° are generally considered hydrophilic, and those with an angle greater than 90° are considered hydrophobic. Surfaces with a contact angle of more than 150° and sliding angle of less than 10° are considered superhydrophobic [16,17,18]. However, according to the Cassie–Baxter model, upon reaching a stable superhydrophobic state, the surface is characterized by minimal contact with water droplets owing to air bubbles trapped between the liquid and solid body, which makes it difficult for corrosive agents to contact and reduces the adhesion of contaminants to the material surface. Thus, superhydrophobicity underlies the creation of self-cleaning materials and provides anti-corrosion and anti-biofouling functions [19,20,21,22,23,24,25,26].

Imparting superhydrophobic properties to the substrate surface requires a combination of multimodal roughness and a continuous stable layer of hydrophobic agents that provide low surface energy at the interface [27,28,29]. The Young equation shows that the surface energy of the substrate is the greatest contributor to the increase in hydrophobic properties for the same surface topology; the lower the surface energy of the substrate, the greater the contact angle [30]. The lowest surface energy is known to correspond to compounds with alkyl and fluoroalkyl substituents [31,32]. For example, for polytetrafluoroethylene, it is approximately 19–21 mN/m [33,34], and for perfluoroalkylacrylates (with long perfluorinated substituents), it is approximately 10–12 mN/m [35]. Polyhedral oligomeric silsesquioxanes are noteworthy and are characterized by extremely low values of surface energy (in the order of 10 mN/m) and chemical resistance to a wide range of solvents [36,37]. However, the practical use of these compounds is not economically viable. Alkylsilanes [38], fluoroalkylsilanes [39,40], and fatty carboxylic acids [41] have been successfully used as low surface energy agents to create superhydrophobic surfaces.

Fluorocarbon polymer coatings are of particular interest for superhydrophobic surfaces due to the extremely low surface energy of fluorine-containing (CF_2_ and CF_3_) functional groups [42]. Compounds of this type include polymers based on silanes and methacrylates with alkyl and perfluoroalkyl substituents [43,44,45,46]. Their use is mainly limited by poor adhesion to the substrate surface. Therefore, polymeric modifiers, which provide a decrease in surface energy but are also capable of covalent fixation on the substrate surface, are of paramount interest.

Previously, our research team proposed the use of copolymers based on fluoroalkyl methacrylate (FMA) and glycidyl methacrylate (GMA) containing reactive epoxy groups as agents to reduce the surface free energy to modify aluminum surfaces and cellulose-containing materials [47,48,49]. Grafting these polymeric modifiers onto a pre-textured aluminum surface provides a superhydrophobic state with contact angles up to 169°. However, special attention should be paid not only to achieving a superhydrophobic state but also to the effect of the copolymer composition and content of fluoroalkyl groups on the modified surface on the free energy and the water droplet sliding angle. These parameters are essential for achieving stable heterogeneous wetting of superhydrophobic coatings and to evaluate the prospects of using GMA and FMA copolymers for creating self-cleaning materials. Thus, this study aimed to study the effect of the composition of reactive GMA and FMA copolymers on the surface free energy and lyophilic properties of polymer coatings based on these copolymers.

## 2. Materials and Methods

### 2.1. Materials

The surface energy studies used 20 × 10 mm microscope slides. Roll-off angle studies used A5 aluminum samples with a size of 20 × 10 mm and a thickness of 0.8 mm, and 38% hydrochloric acid, and solvents, such as methyl ethyl ketone (MEK), n-hexane, deionized water, n-decane, and diiodomethane were purchased from Vekton (Russia).

Glycidyl methacrylate (GMA, 97%), 2,2,2-trifluoroethyl methacrylate (TEMA, 99%), 1,1,1,3,3,3-hexafluoroisopropyl methacrylate (HIMA, 99%), 2,2,3,4,4,4-hexafluorobutyl methacrylate (HFMA, 98%), 2,2,3,3,4,4,4-heptafluorobutyl methacrylate (HBMA, 99.5%), and azobisisobutyronitrile (AIBN, 98%) were purchased from Sigma-Aldrich.

Glycidyl methacrylate was vacuum distilled at 50 °C before use.

### 2.2. Synthesis of Random GMA and FMA Copolymers and Modification of Materials

Random copolymers based on GMA and FMA were synthesized in MEK (with molar ratios of monomers GMA/FMA = 2:1, 1:1, and 1:2.2) at 70°C for 24 h, with a total monomer concentration of 1 mol/L. Azobisisobutyronitrile (AIBN) was used as an initiator. GMA with FMA copolymers was precipitated in cold hexane, followed by drying under reduced pressure until constant weight.

Copolymer solutions were prepared in MEK according to the previously described procedure [47].

### 2.3. Attachment of Synthesized Copolymers to the Glass Surface

The glass samples were pre-washed with a soapy solution and then with distilled water, followed by drying in an oven at 80°C for 30 min. Then, the dried glass samples were treated with low-pressure oxygen plasma using the Diener–Femto low-pressure plasma system (Germany) at an operating pressure of 0.3 mbar for 15 min. The cleaned samples were immersed in 3 wt.% FMA and GMA copolymer solutions for 1 h. Then, the glass samples were extracted from the modifier solutions, air dried for 1 min to remove the solvent from the surface, and then placed in Petri dishes for further heat treatment at 140°C for 1 h.

### 2.4. Attachment of Synthesized Copolymers to the Textured Aluminum Surface

The aluminum surface was pre-cleaned using the method described in [50]. The aluminum surface was textured by etching with a 5 M hydrochloric acid solution. Acid and etch products were washed off by boiling in deionized water. Then, the aluminum samples were placed in a drying oven at 140 °C for 40 min. 

The aluminum surface was modified with FMA and GMA copolymers using the abovementioned procedure for glass samples.

### 2.5. Methods

The morphological features and surface chemical composition of the modified aluminum samples were investigated with scanning electron microscopy (SEM) using a Versa 3D device (FEI, Hillsboro, OR, USA) equipped with an EDAX Apollo X energy dispersive (EDS) microanalyzer with integrated Team software N 4.6.1000.0285 (Warrendale, PA, USA) in low vacuum mode at a water vapor pressure in the chamber of 10–80 Pa with an accelerating voltage of 15 to 20 kV and beam current of 13 pA to 4 nA.

Elemental analysis of the polymers was performed on the CHNOS elemental analyzer “Vario EL Cube” (Elementar, Langenselbold, Germany) using the “2 mg 70 s” method. The analysis time for one sample was 10 min, He consumption was 230 mL/min, and O_2_ consumption was 38 mL/min with an oxygen supply time of 70 s. The temperatures of the oxidation and reduction columns were 1150 and 850 °C, respectively.

The contact angle of wetting and surface energy were determined using the DataPhysics OCA 15 EC system (DataPhysics, Filderstadt, Germany) with integrated SCA 20 software for calculating the surface free energy and a database containing the surface tension of various liquids. The measurements were performed by applying drops of deionized water, n-decane, and diiodomethane with a volume of 5–7 μL on the surface of the substrate, and the contact angle of a sessile drop was calculated according to the Young–Laplace method. Six to eight measurements were performed, and the arithmetic mean of the contact angles was calculated.

Dynamic studies of the drop behavior on the surface of the modified samples at long time intervals were carried out in a cell saturated with water vapor. Because a reduced drop evaporation rate was observed on the modified surface under conditions of high humidity and lack of contact with the external environment, this enabled us to study the changes in the contact angle of a sessile drop at long time intervals. Contact angle measurements were carried out in accordance with the procedure described above.

The roll-off angle was determined using the DataPhysics OCA 25 system (DataPhysics, Filderstadt, Germany) with a TBU100 tilting base unit. The measurements were performed in two ways. The first method consisted of dropping 6 μL of deionized water onto the sample surface, followed by tilting the surface by varying the angular velocity (from 0.37 to 1.1°/s). The roll-off angle is the minimum angle of inclination of the sample that enables spontaneous rolling of a water droplet off the surface. Each sample was subjected to at least 7 measurements with the calculation of the arithmetic mean value. The second method of measuring the roll-off angle involved the following experiment. The sample was placed on a table with a certain angle of inclination (3, 5, and 7°), and 6 μL of deionized water was dropped onto its surface. The droplet was detached from the end of the needle to eliminate the effect of the momentum of the falling droplet. The experiment was repeated 40 times covering the entire surface of the sample, and the number of droplets that rolled down and remained on the surface was recorded. Then, the percentage of droplets that rolled down at a given angle of inclination was calculated.

The Owens–Wendt–Rabel–Kaelble (OWRK) method, which is a standard procedure applied to hydrophobic materials to measure the contact angles of wetting involving at least two liquids, was used to calculate the surface free energy (SFE). The OWRK method is applicable to all polymer coatings and can be used to determine the dispersion and polar contributions of solid–liquid interactions. The calculation was performed using DataPhysics SCA 20 software.

The wetting angles of the surface modified with copolymers were determined using three liquids with different values for the dispersive and polar parts of the surface tension. Deionized water was used as a polar liquid, and diiodomethane and n-decane were used as dispersion liquids.

The work of adhesion, i.e., the work expended on overcoming the adhesive forces during the separation of particles of two dissimilar surfaces, was determined in terms of contact angle and surface tension using the Young–Dupré equation:(1)$$W_{sl}=\gamma_{l}\left(1+\cos\theta \right),$$
where *γ_l_* is the surface tension of the wetting liquid in mN/m and cos*θ* is the cosine of the contact angle.

The adhesive energy between a solid and a liquid can be separated into interactions between the dispersive and polar parts of the two phases using the equation [51]:(2)$$W_{sl}=2\left[{\left(\gamma_{l}^{d}\right)}^{\frac{1}{2}}{\left(\gamma_{s}^{d}\right)}^{\frac{1}{2}}+{\left(\gamma_{l}^{p}\right)}^{\frac{1}{2}}{\left(\gamma_{s}^{p}\right)}^{\frac{1}{2}}\right],$$
where $\gamma_{l}^{d}$ and $\gamma_{s}^{d}$ are the dispersive parts and $\;\gamma_{l}^{p}$ and $\gamma_{s}^{p}$ are the polar parts for the liquid and the solid surface, respectively. This equation is used in the DataPhysics SCA 20 software.

## 3. Results and Discussion

The water-repellent properties of the substrate were determined by the chemical composition of the surface and build-up depending on the multimodal roughness of the surface layer. The wettability of the substrate surface was assessed by measuring the contact angle, which can be up to 120° on a flat surface. However, regardless of the surface microtexture, the change in hydrophobic properties can only be controlled by varying the chemical composition of the surface. Therefore, the surface free energy is the critical parameter that determines the potential use of polymeric modifiers for creating water-repellent and self-cleaning coatings because it provides hydrophobic properties and reduces the adhesion of contaminants to the material surface.

Modern “green chemistry” trends are aimed at abandoning perfluorinated compounds [52,53]. Therefore, we propose to use a number of copolymers based on glycidyl methacrylate and fluoroalkyl methacrylates with a low fluorine content in the monomer unit, which are promising agents for reducing surface free energy and are not inferior to perfluorinated compounds. Fluorinated substituents reduce surface energy, and glycidyl groups provide covalent fixation of the modifier on the substrate surface [49,54,55]. In addition, we are interested in studying the effect of the acrylate monomer unit structure with a low fluorine content ranging from three to seven atoms and the copolymer composition with varying contents of the anchor and functional comonomers on the change in surface energy; we are also interested in comparing the hydrophobic properties of homo- and copolymers of fluoroalkyl methacrylates.

The composition of the synthesized FMA and GMA copolymers was confirmed by elemental analysis. The experimental data (Table 1) on the carbon and hydrogen content showed that the resulting ratio of [FMA]/[GMA] comonomers was near the theoretical value. However, the observed increased content of glycidyl methacrylate in the copolymers was due to the peculiarities of the copolymerization of these monomers. Because the deviation from the theoretical composition was insignificant and typical for all pairs of comonomers, this study used the following [FMA]/[GMA] ratios to designate the composition: 1:2 (33.3% FMA), 1:1 (50% FMA), 1:0.5 (66.6% FMA), and 1:0 (100% FMA).

The contact angle depends on the liquid used for the measurements and can be used to determine the wettability of the substrate surface and evaluate the SFE of a solid body [56]. This study used the OWRK method to determine the SFE, which is a universal technique for polymer coatings that implies separation of the SFE into polar and dispersive parts. The dispersive part refers to the weak dispersion forces (van der Waals forces), and the polar part is associated with all non-dispersion forces: hydrogen bonds, Coulomb interactions, dipole interactions, and acid–base interactions.

Smooth surfaces are traditionally used to measure surface energy; therefore, polymer-coated mineral glass was used as a model substrate to study the effect of the structure and composition of the FMA and GMA copolymer on the SFE. Contact angles were determined using three liquids with different values for the dispersive and polar parts of the surface tension. Water was used as a polar liquid, and diiodomethane and n-decane were used as dispersion liquids.

Table 2 shows that the grafting of GMA- and FMA-based copolymers can reduce the surface free energy by more than three times as opposed to the poly-GMA-based homopolymer coating, along with an increase in contact angles for all test liquids. For water, the maximum wetting angle on a smooth surface modified with poly-GBMA homopolymer was 111°. The minimum content of FMA in the copolymer (33%) provides a reduction in SFE from 1.5 to 2.5 times. Increasing the FMA functional comonomer content to 66% brings the total SFE value of the copolymers closer to that of the FMA homopolymers. For example, the difference between a poly-(HBMA-co-GMA) copolymer and a poly-(HBMA) homopolymer is only 0.06 mN/m. Assuming the same comonomer ratios, an increase in the fluorine content in the elementary unit from three to seven atoms significantly contributes to the reduction in the surface energy. Thus, the difference in SFE for similar copolymer compositions is approximately 9 mN/m. The lowest SFE value (equal to 13.76 mN/m) is representative of the polymer coating based on the poly-(HBMA-co-GMA) copolymer, which is comparable to the SFE values for polyhedral oligomeric silsesquioxanes and perfluorinated acids [35,36,37].

Despite the six fluorine atoms in the monomer unit, the poly-(HFMA-co-GMA) copolymer is characterized by SFE values that are similar to those of poly-(TEMA-co-GMA) with three fluorine atoms. This feature occurs because the structure of the fluoroalkyl substituent of the HFMA monomer has one hydrogen atom on the third carbon, which is not substituted by a fluorine atom. This leads to an increase in the relative electronegativity of fluorine atoms due to incomplete substitution of fluorine on one of the carbons, which enhances the proneness to wetting by polar liquids capable of forming hydrogen bonds; this, in turn, increases the polar part of the SFE [57]. Therefore, poly-(HFMA-co-GMA)-based polymer coatings are not mentioned in further discussion in the analysis of superhydrophobic properties and their stability.

In determining the SFE using three test liquids, the estimated value of the correlation coefficient is more than 90% for all modifiers, which indicates the reproducibility of results. The estimated SFE values for the poly-TEMA-based homopolymer coatings agree with the literature data [33,35].

The dispersive part is the main contributor to the surface free energy of the FMA-based coatings and thus accounts for 96–99% of the total surface energy. However, for the poly-GMA homopolymer, the polar part has a much higher value of 10.81 mN/m and thus accounts for 25% of the total surface energy. Therefore, the proposed hydrophobic coatings are not wetted by polar liquids, whereas the initial substrate and the one modified with the poly-GMA homopolymer have a hydrophilic nature. Figure 1 shows the dependences of the work of adhesion of a polar liquid (water) and two non-polar liquids (diiodomethane and n-decane) on the surface of polymer coatings with varying FMA content in the copolymer. The work of adhesion was calculated using Equations (1) and (2). Their values are similar; therefore, we used data obtained by the Young–Dupré equation. More information can be found in Appendix A. For all coatings of interest, the greatest effect of adhesion corresponds to diiodomethane and water. The number of fluorine atoms in the monomer unit is the largest contributor to the decrease in the work of adhesion. Thus, assuming the same FMA content in the copolymer, the difference in the work of adhesion for poly-(TEMA-co-GMA) and poly-(HBMA-co-GMA) copolymers is approximately 15 mN/m. Given an increase in the [FMA] content to 66%, the observed values of the work of adhesion tend to reach the values representative of homopolymers.

A hierarchical structure of the near-surface substrate layer is an indispensable condition for studying the effect of the composition of copolymers on superhydrophobic properties and the self-cleaning capacity. Therefore, aluminum pre-textured by chemical etching in hydrochloric acid was used as a reference substrate. Figure 2 shows that the processed aluminum surface is characterized by multimodal roughness, consisting of a combination of microprotrusions with a cellular nanostructure. Analysis of the structure and peculiarities of the attachment of GMA- and FMA-based copolymers on the aluminum surface was considered in depth in previous publications [47,49].

The chemical composition of polymer coatings on the textured aluminum surface was determined by energy dispersive analysis, which can record elements on the substrate surface to a depth of several micrometers. Table 3 shows that the microtexture obtained by etching consists mainly of aluminum oxy-forms, with the oxygen concentration constituting 7.1 at.%. Fluorine, the concentration of which increases with an increase in the FMA content in the copolymer, serves as the principal indicator element of the grafting of the polymer coating based on FMA and GMA copolymers. Thus, an average increase in the FMA content to 66% led to a 1.6-fold change in the concentration of fluorine atoms, while the same contribution resulted from an increase in the fluorine content in the monomer unit. The highest concentration of fluorine on the surface was detected for samples modified with FMA homopolymers, which resulted in the absence of the contribution of carbon and oxygen from GMA. However, FMA-based homopolymers are attached to the surface only through physical interactions; therefore, of particular interest is the study of the effect of the GMA anchor comonomer on the stability of the superhydrophobic state.

The calculation of the SFE of superhydrophobic coatings yields extremely low values (Table 2) [58,59]. This is due to an increase in the wetting angles used for the calculation owing to the multilevel roughness of the near-surface layer of the material. The free energy of the coating surface based on the poly-GMA homopolymer on a smooth substrate is 40.24 mN/m (Table 2). Table 4 shows that the contact angles of water droplets on the poly-GMA of a textured substrate have rather high values of 144°, but this is not enough to impart superhydrophobicity to the surface. The obtained results demonstrate that the low SFE of polymer coatings based on FMA and GMA copolymers on a smooth substrate (from 25 to 13 mN/m) provides a textured surface with superhydrophobic properties and contact angles from 159° to 170°. Poly-(TEMA-co-GMA) shows the greatest effect of the copolymer composition on the contact angle. Contact angles up to 168° corresponding to the wetting of the poly-TEMA homopolymer-based coating were observed only when the TEMA content in the copolymer reached 66%. For copolymers of poly-(HIMA-co-GMA) and poly-(HBMA-co-GMA), the copolymer composition had no effect on the initial contact angles, which were in the range 166–170°.

When comparing the lyophilic properties and the work of adhesion (data on the work of adhesion can be found in Appendix A), based on changes in the copolymer composition, the parameters of interest change stepwise on the smooth surface (Figure 3a,c). With a decrease in the SFE, and with the appearance of roughness (Figure 3b,d) of the near-surface layer, the smooth change of the results occurs. Regardless of the amount of FMA in the copolymer, the SFE values tend to be near the values representative of homopolymers with an increase in fluorine in the monomer unit ranging from three to seven atoms (Figure 3d). This results in a significant increase in the stability of the superhydrophobic properties of the system in terms of the contact angles and the work of adhesion.

However, the achievement of high contact angles does not allow us to conclude that the superhydrophobic properties of coatings are stable [60]. The instability of the superhydrophobic state manifests itself as a change in the wetting regime from heterogeneous to homogeneous, which coincides with a significant decrease in the wetting angle. Assuming a long contact time, a decrease in the wetting angle may be due to the penetration of water into the rough surface or interaction with oxygen-containing groups in defective areas of the coating with the formation of a new wetting surface.

Figure 4 shows that the stability of the superhydrophobic state increases with an increase in the FMA content of the copolymer and the number of fluorine atoms in the FMA monomer units. For all copolymers, the highest stability was observed at an FMA content of 66%. Thus, for poly-(TEMA-co-GMA), the contact angle was 157°, and for poly-(HBMA-co-GMA), the contact angle was 164° after 25 h of contact.

GMA, which is part of the copolymers, provides an anchoring bond and is essential for the covalent attachment of the modifier to the surface. Meanwhile, a decrease in the FMA content provides a cheaper coating and a sufficiently low SFE. However, an increase in the GMA content of the copolymer results in an increase in the number of oxygen-containing functional groups, resulting in the adsorption of water molecules and the formation of hydrogen bonds. The lack of fluorinated substituents, which also perform a screening function, affects the stability of the superhydrophobic properties of the coatings. Given the FMA content of 33%, the composition of poly-(TEMA-co-GMA) features a sharp decrease in contact angles to 130° within 5 h of contact (Figure 4a), which indicates a transition to a homogeneous wetting regime.

These considerations are visualized as a dependence of the wetting angle change during contact as a function of the free energy of the polymer coating (Figure 5). With a decrease in the SFE for a polymer coating based on GMA and FMA copolymers, the absolute value between the initial and final wetting angles tends to decrease under conditions of prolonged contact of a water droplet with the surface. However, this dependence is representative only of copolymers with a capacity for chemical sealing on the surface. Figure 4 and Figure 5 (points 4) show that the change in contact angles over time as an indicator of the stability of superhydrophobic properties of the polymer coatings based on FMA homopolymers does not correlate with the corresponding lowest SFE values. This occurs due to the attachment of FMA homopolymers to the surface only via physical interactions, which enables the processes of macromolecular rearrangements and desorption of the modifier from the substrate surface, resulting in deterioration of the hydrophobic properties of the coating [61].

In addition to the contact angle, the superhydrophobic properties of the surface are characterized by another vital and “practical” parameter, i.e., the roll-off angle, which is the minimum angle of inclination at which droplets spontaneously roll off the surface. This parameter is useful for evaluating the ease of rolling and characterizing the self-cleaning properties of a surface. In some cases, high wetting angles correspond to high roll-off angles, which indicates either defects in the superhydrophobic coating and a transitional (unstable) wetting state or topological features of the surface [30]. However, we have noticed scattered knowledge regarding the methods for measuring the roll-off angle in the literature. For example, the authors of [62] used a “home-made instrument” to measure the roll-off angle without providing a description of the device and the detection method. The authors of [63] performed measurements using a commercial device (SA-11, Kyowa); unfortunately, there is no description of the experimental procedure. The authors of [64,65], who provide the most complete description of the methodology for measuring roll-off angles, indicate the following parameters: the method of dropping and the volume of the tested water droplets, the rate of inclination of the surface to the horizon, and the number of parallel measurements.

To put this into perspective, the roll-off angles were measured using two methods. The first technique is aimed at studying the effect of the surface free energy of polymer coatings on the roll-off angle with changing rate of inclination of the sample surface. The measurements of the rate of inclination were performed in the range from 0.37 to 1.1°/s. The data in Table 5 show that at a uniform angular velocity, all modified aluminum samples are characterized by low roll-off angles from 10.8° to 2.3°. The values of the contact angle practically do not differ with a decrease in the SFE due to an increase in the FMA content of the copolymer or the amount of fluorine in the monomer unit, but the roll-off angle noticeably decreases (Table 5, Figure 6), which is a positive factor for applying those compositions for creating self-cleaning polymer coatings.

An increase in the rate of inclination significantly affected the roll-off angles from the surface modified with a polymer coating with the same SFE. This occurred because surface tilting is a circular motion, and the moment of inertia and the kinetic energy of the droplet change as a function of the given angular velocity. The study used 6-μL droplets; therefore, it can be assumed that they have the same mass and spherical shape because the initial contact angles are almost the same. The kinetic energy of the system during rotational motion (when the surface is tilted relative to the instrument axis) is proportional to the square of the angular velocity ($E_{kin}=\frac{L\omega^{2}}{2}$, where L is the moment of inertia and ω is the angular velocity). It is possible to trace the change in kinetic energy with increasing angular velocity: $\frac{E_{\omega 2}}{E_{\omega 1}}=2.7$ and $\frac{E_{\omega 3}}{E_{\omega 1}}=8.8$. This result shows that an increase in the angular velocity by less than 1°/s increases the kinetic energy by more than 8 times and contributes to the change in roll-off angles. Figure 6 shows that for coatings with the same SFE, an increase in the angular velocity from 0.37 to 1.1°/s results in a noticeable decrease in roll-off angles. Thus, the change is approximately 2° for poly-(TEMA-co-GMA) and poly-(HBMA-co-GMA). An increase in the rate of inclination allows control of the roll-off angle, which is an advantage for creating coatings on tilted surfaces but may cause misunderstanding when presenting experimental data by disregarding this parameter. Therefore, we recommend measuring the roll-off angle at a low angular velocity for correct presentation of the results.

The second method is based on the statistics of water droplets rolling off a surface with a preset slope. Table 6 shows that the percentage of the rolled off droplets naturally increases with an increase in the inclination angle and is already more than 90% at 5°. The wetting results based on both methods of measuring the roll-off angle on the aluminum surface modified with GMA- and FMA-based copolymers allowed us to conclude that a stable superhydrophobic state is characterized by high wetting angles up to 170° and low roll-off angles of less than 5°.

The self-cleaning phenomenon is one of the main characteristics for the practical application of superhydrophobic materials. Figure 7 shows that water droplets rolling off the sample surface capture soil particles, and the surface remains almost clean after rolling off of a few droplets (video can be found in the Appendix A). However, deceleration of a water droplet was observed when it interacted with the surface of a soil particle (Figure 7a–f); hence, the volume of the droplet and the angle of inclination are the main contributors to this effect. This observation emphasizes the need to measure roll-off angles as the principal property of superhydrophobic materials.

Figure 8 shows the self-cleaning effect of superhydrophobic aluminum (without inclination to the horizon) due to the collection of soil particles by the movement of a water droplet over the surface (video can be found in the Appendix A). The above photographs show that the soil particles have no affinity for surfaces with low surface energy and are easily collected by a single water droplet with a significant increase in its volume. In addition, the impact of a water jet and large water droplets with high kinetic energy were used to visualize the self-cleaning effect from an aclinal surface. However, water droplets bounced off the superhydrophobic surface and soil particles moved to the liquid–vapor interface (Appendix A; video can be found in the Appendix A). Thus, modification of the textured aluminum surface with GMA- and FMA-based copolymers with low surface energy provides a superhydrophobic state and a self-cleaning effect.

## 4. Conclusions

In this study of polymer coatings based on several glycidyl methacrylate and fluoroalkyl methacrylate copolymers with different numbers of fluorine atoms in the monomer unit ranging from three to seven and different contents of the hydrophobic comonomer, the effect of the SFE on the lyophilic properties of the modified substrates is shown, based on elemental analysis and EDS methods. Polymer coatings based on FMA and GMA copolymers provide a low SFE. On a smooth substrate, the SFE values are in the range of 25 to 13 mN/m, which provides superhydrophobic properties on the textured aluminum surface with contact angles ranging from 159° to 170° and increases the stability of the superhydrophobic state upon prolonged contact with water. The lowest SFE (equal to 13.76 mN/m) is representative of a polymer coating based on the poly-(HBMA-co-GMA) copolymer, which is comparable to the SFE values for polyhedral oligomeric silsesquioxanes and perfluorinated acids. A decrease in the SFE with an increase in the amount of fluorine in the monomer unit ranging from three to seven atoms and the FMA content of the copolymer had almost no effect on the initial wetting angles on the textured aluminum surface; however, the rolling off angle noticeably decreased from 10.8° to 2.3°, which demonstrates that these compositions can be used in the creation of self-cleaning polymer coatings.

## Figures and Tables

**Figure 1 polymers-14-01960-f001:**
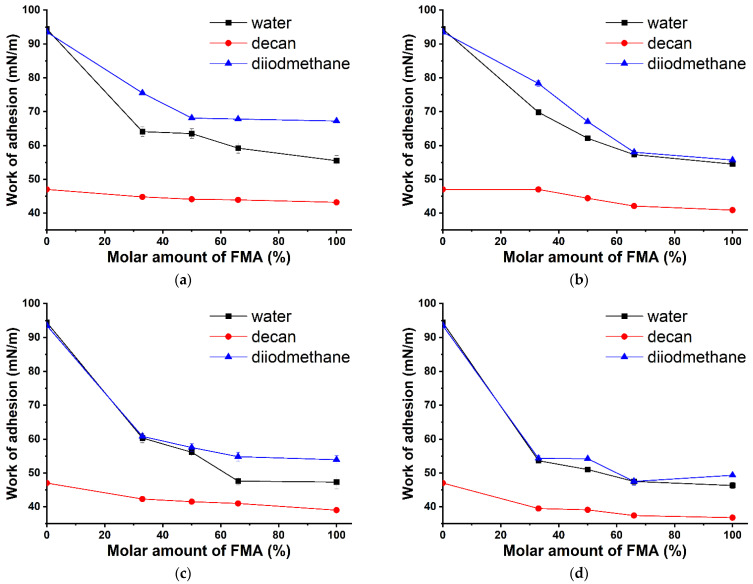
Change in the work of adhesion of the test liquids (water, n-decane, and diiodomethane) on the surface of glasses modified with FMA- and GMA-based copolymers as a function of the FMA content in the copolymer: (**a**) poly-(TEMA-co-GMA); (**b**) poly-(HFMA-co-GMA); (**c**) poly-(HIMA-co-GMA); (**d**) poly-(HBMA-co-GMA).

**Figure 2 polymers-14-01960-f002:**
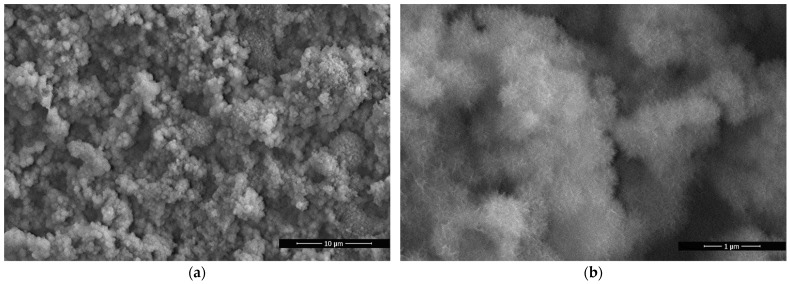
SEM images of the textured aluminum surface modified with poly-(HBMA-co-GMA): (**a**) 8000×; (**b**) 60,000×.

**Figure 3 polymers-14-01960-f003:**
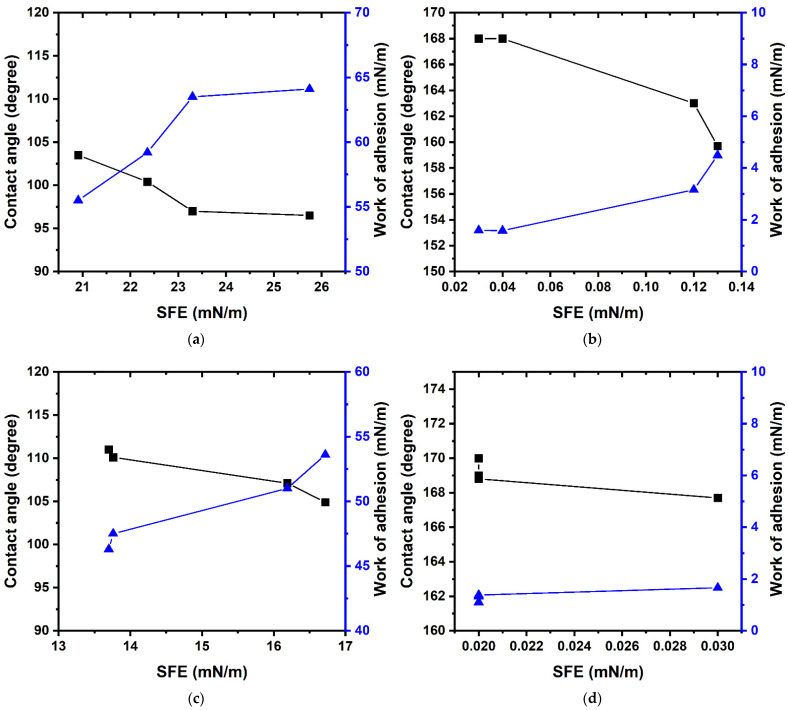
Dependence of the contact angle and the work of adhesion on the surface free energy for grafted coatings based on the poly-(TEMA-co-GMA) ((**a**)—glass; (**b**)—textured aluminum) and poly-(HBMA-co-GMA) copolymers ((**c**)—glass; (**d**)—textured aluminum).

**Figure 4 polymers-14-01960-f004:**
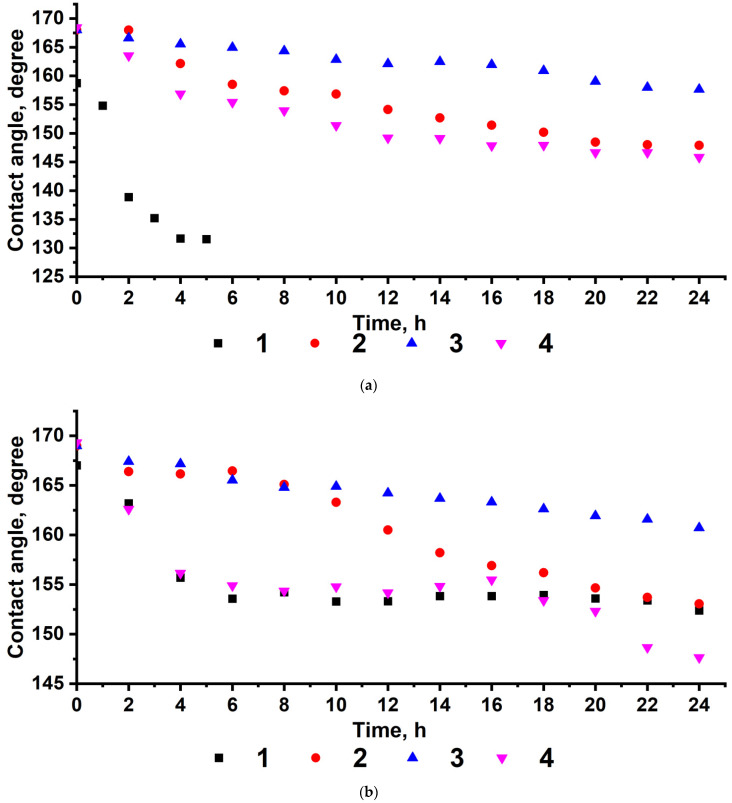
Change in wetting angles from the time of contact of a water droplet with the textured aluminum surface modified with (**a**) poly-(TEMA-co-GMA); (**b**) poly-(HIMA-co-GMA); and (**c**) poly-(HBMA-co-GMA) with varying FMA contents of the copolymer (theoretical FMA contents are as follows: 1—33%; 2—50%; 3—66%; 4—100%).

**Figure 5 polymers-14-01960-f005:**
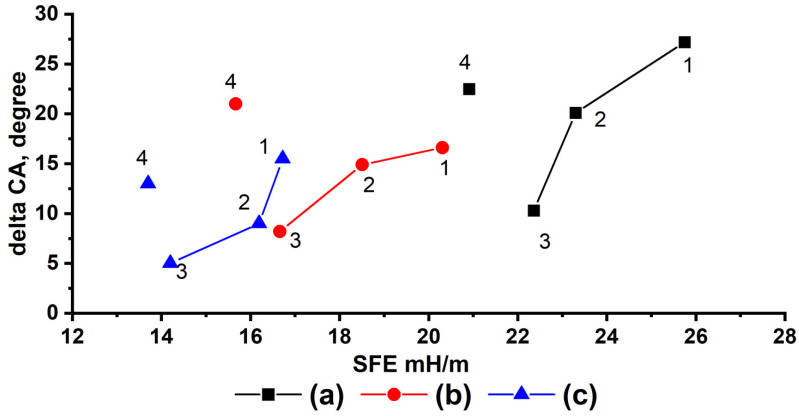
Dependence of the change of wetting angle after 24 h of contact of a water droplet with the textured aluminum surface modified with (**a**) poly-(TEMA-co-GMA); (**b**) poly-(HIMA-co-GMA); and (**c**) poly-(HBMA-co-GMA) from the surface free energy with varying FMA contents of the copolymer (1—33%; 2—50%; 3—66%; 4—100%).

**Figure 6 polymers-14-01960-f006:**
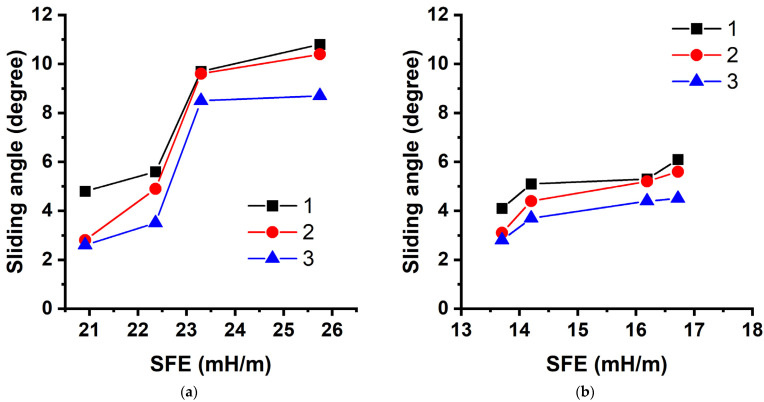
Changes in the roll-off angles of water droplets from the surface of the textured aluminum samples as a function of the surface free energy of the polymer coating at different rates of inclination: (**a**) poly-(TEMA-co-GMA); (**b**) poly-(HBMA-co-GMA): 1—0.37°/s; 2—0.61°/s; 3—1.1°/s.

**Figure 7 polymers-14-01960-f007:**
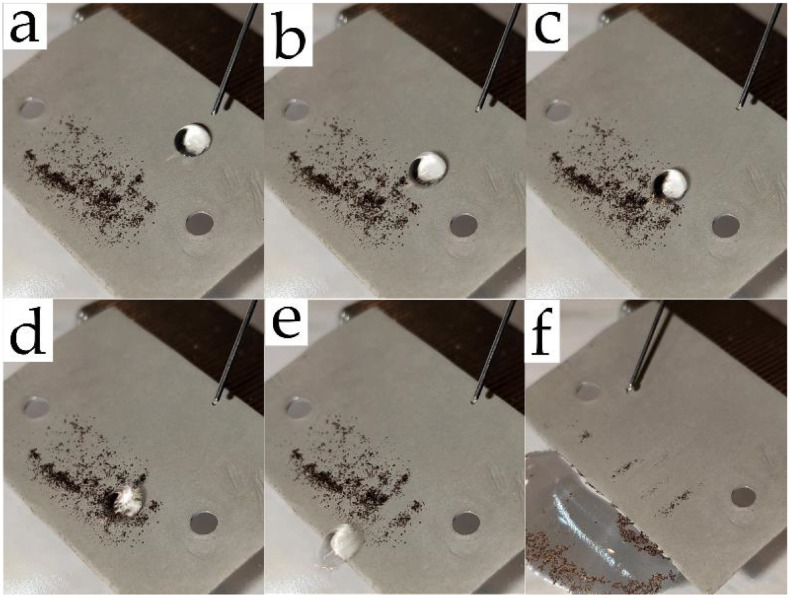
Self-cleaning effect due to the rolling off of a water droplet from a tilted plane (10°) of a superhydrophobic aluminum surface at various points in time: (**a**) 0; (**b**) 50; (**c**) 80; (**d**) 100; (**e**) 170 ms; (**f**) surface after rolling off of 20 water droplets.

**Figure 8 polymers-14-01960-f008:**
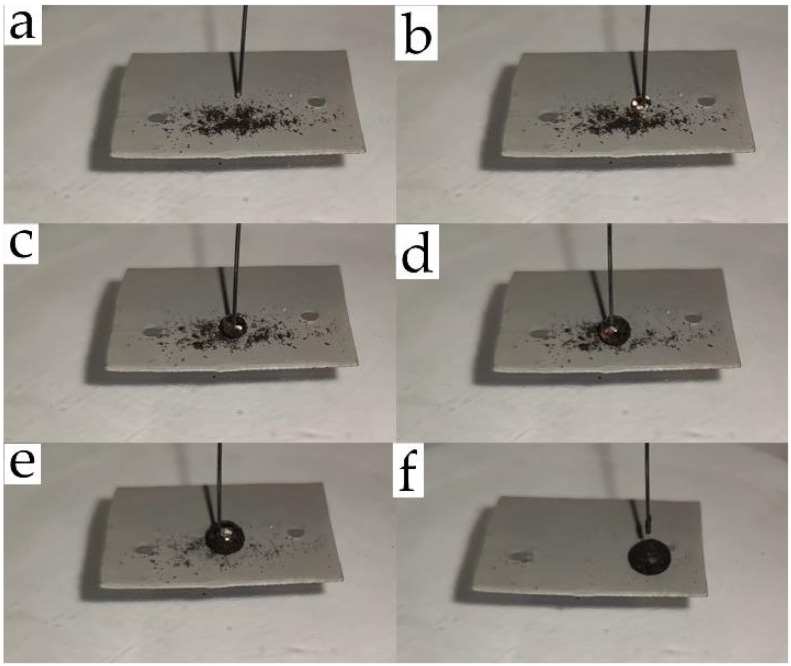
Demonstration of the self-cleaning effect as a result of the collection of soil particles from the superhydrophobic aluminum surface due to the movement of a water droplet across the surface: (**a**) 0; (**b**) 3; (**c**) 6; (**d**) 9; (**e**) 18; (**f**) 45 s.

**Table 1 polymers-14-01960-t001:** Results of the elemental analysis of FMA and GMA copolymers (using the CHNOS method).

Copolymer	Molar Content [FMA], %		Elemental Content, %	
	Theoretical	Experimental	C, %	H, %
Poly-(TEMA-co-GMA)	33.3	29.0	53.80	6.55
	50.0	44.4	51.19	5.75
	66.7	62.1	48.35	4.50
Poly-(HFMA-co-GMA)	33.3	29.0	50.50	6.08
	50.0	41.7	47.96	4.76
	66.7	61.7	43.80	3.78
Poly-(HIMA-co-GMA)	33.3	29.8	49.50	6.13
	50.0	42.7	46.09	4.69
	66.7	64.9	41.37	3.08
Poly-(HBMA-co-GMA)	33.3	29.3	48.75	4.89
	50.0	46.7	44.56	3.94
	66.7	65.8	40.82	3.22

**Table 2 polymers-14-01960-t002:** Initial contact angles and surface energy of glass samples modified with GMA and FMA copolymers.

Modifier	Molar Content [FMA], %	Contact Angle of Wetting, °			SE, mN/m	D, mN/m	*p*, mN/m	RQ *
		Water	n-Decane	Diiodomethane				
Poly-GMA	0	72.2 ± 1.1	14.0 ± 0.9	32.7 ± 2.1	40.24 ± 0.84	29.49 ± 0.38	10.75 ± 0.47	0.95
Poly-(TEMA-co-GMA)	29.0	96.5 ± 1.0	28.6 ± 1.0	61.0 ± 0.4	25.75 ± 0.41	23.38 ± 0.15	2.36 ± 0.25	0.98
	44.4	97.0 ± 1.5	31.7 ± 0.5	70.4 ± 0.3	23.30 ± 0.43	20.55 ± 0.13	2.75 ± 0.38	0.99
	62.1	100.4 ± 1.4	32.6 ± 1.0	71.1 ± 0.6	22.36 ± 0.45	20.47 ± 0.18	1.89 ± 0.27	0.99
Poly-TEMA	100	103.5 ± 1.4	35.5 ± 0.4	74.1 ± 0.4	20.91 ± 0.33	19.53 ± 0.14	1.38 ± 0.22	0.99
Poly-(HFMA-co-GMA)	29.0	92.0 ± 1.0	13.0 ± 0.3	57.3 ± 1.1	28.59 ± 0.42	25.33 ± 0.26	3.25 ± 0.30	0.99
	41.7	98.1 ± 1.0	30.3 ± 0.4	71.4 ± 0.6	22.97 ± 0.32	20.52 ± 016	2.46 ± 0.19	0.99
	61.7	101.9 ± 0.7	39.8 ± 0.7	81.8 ± 0.7	19.08 ± 0.35	16.91 ± 0.14	2.17 ± 0.16	0.96
Poly-HFMA	100	104.2 ± 0.4	44.2 ± 0.7	84.5 ± 1.0	17.65 ± 0.52	15.79 ± 0.41	1.86 ± 0.11	0.94
Poly-(HIMA-co-GMA)	29.8	99.5 ± 1.3	39.3 ± 2.5	78.6 ± 2.3	20.31 ± 0.68	17.65 ± 0.84	2.66 ± 0.14	0.99
	42.7	103.0 ± 1.0	42.3 ± 0.6	82.4 ± 1.2	18.51 ± 0.49	16.49 ± 0.40	2.02 ± 0.17	0.96
	64.9	109.9 ± 1.0	44.1 ± 0.7	86.1 ± 1.8	16.66 ± 0.69	15.88 ± 0.64	0.79 ± 0.10	0.92
Poly-HIMA	100	110.3 ± 1.6	50.5 ± 0.5	86.5 ± 1.7	15.67 ± 0.44	14.77 ± 0.40	0.90 ± 0.27	0.97
Poly-(HBMA-co-GMA)	29.3	104.9 ± 0.6	49.0 ± 1.4	86.0 ± 0.8	16.72 ± 0.36	14.83 ± 0.38	1.90 ± 0.40	0.95
	46.7	107.1 ± 0.5	50.2 ± 0.5	86.1 ± 0.4	16.19 ± 0.32	14.73 ± 0.23	1.46 ± 0.10	0.96
	65.8	110.1 ± 0.8	55.2 ± 0.6	93.7 ± 1.2	13.76 ± 0.42	12.46 ± 0.35	1.29 ± 0.09	0.90
Poly-HBMA	100	111.0 ± 0.6	57.0 ± 0.6	91.6 ± 0.3	13.70 ± 0.32	12.74 ± 0.13	0.96 ± 0.20	0.91

RQ * is a correlation coefficient.

**Table 3 polymers-14-01960-t003:** Chemical composition of the initial and modified aluminum surface (using the EDS method).

Modifier	Molar Content [FMA], %	Concentration, At.%			
		Al	O	C	F
Initial Al	---	91.9	3.1	4.9	---
Textured Al	---	92.4	7.1		---
Poly-GMA	0	17.3	50.1	32.5	---
Poly-(TEMA-co-GMA)	29.0	20.5	54.1	22.4	2.9
	44.4	26.6	52.4	18.1	3
	62.1	23.6	51.7	20.6	4.2
Poly-TEMA	100	22.6	50.9	20.4	6.1
Poly-(HFMA-co-GMA)	29.0	21.3	60.5	13.7	4.5
	41.7	24	51.3	19.8	4.9
	61.7	19.4	49.3	24.8	6.6
Poly-HFMA	100	21.9	62.4	8.4	7.3
Poly-(HIMA-co-GMA)	29.8	23.1	52.4	21.1	3.4
	42.7	23.7	52.8	19	4.5
	64.9	22.6	50.5	21.4	5.5
Poly-HIMA	100	24.2	50	18.2	7.6
Poly-(GMA-co-HBMA)	29.3	27.5	48.9	19.9	3.6
	46.7	23.4	53.2	18	5.3
	65.8	19.7	48.4	25	6.8
Poly-HBMA	100	19.6	51.4	19.9	9.1

**Table 4 polymers-14-01960-t004:** Initial contact angles and surface free energy of textured aluminum modified with GMA- and FMA-based copolymers with different monomer unit contents.

Modifier	Molar Content [FMA], %	Contact Angle of Wetting, °		SE, mN/m	D, mN/m	*p*, mN/m
		Water	Diiodomethane			
Poly-GMA	0	144.2 ± 2	53.5 ± 2	45.89	38.44	7.45
Poly-(TEMA-co-GMA)	29.0	159.7 ± 3	154.6 ± 2	0.13	0.11	0.02
	44.4	163.0 ± 2	154.6 ± 2	0.12	0.12	0.00
	62.1	168.0 ± 3	160.7 ± 3	0.04	0.04	0.00
Poly-TEMA	100	168.0 ± 3	161.9 ± 2	0.03	0.03	0.00
Poly-(HFMA-co-GMA)	29.0	163.8 ± 2	160.7 ± 3	0.05	0.04	0.01
	41.7	165.5 ± 3	161.6 ± 3	0.03	0.03	0.00
	61.7	166.1 ± 3	163.5 ± 3	0.02	0.02	0.00
Poly-HFMA	100	168.7 ± 2	163.6 ± 3	0.02	0.02	0.00
Poly-(HIMA-co-GMA)	29.8	166.3 ± 2	161.1 ± 3	0.04	0.04	0.00
	42.7	167.0 ± 3	162.3 ± 3	0.03	0.03	0.00
	64.9	169.0 ± 3	163.4 ± 3	0.02	0.02	0.00
Poly-HIMA	100	169.2 ± 2	163.6 ± 3	0.02	0.02	0.00
Poly-(GMA-co-HBMA)	29.3	167.7 ± 3	162.1 ± 3	0.03	0.03	0.00
	46.7	168.8 ± 3	163.1 ± 3	0.02	0.02	0.00
	65.8	169.0 ± 3	163.8 ± 3	0.02	0.02	0.00
Poly-HBMA	100	170.0 ± 2	163.8 ± 2	0.02	0.02	0.00

**Table 5 polymers-14-01960-t005:** Roll-off angle of water droplets from the aluminum surface modified with GMA- and FMA-based copolymers with a changing rate of inclination of the plane to the horizon.

Modifier	Molar Content [FMA], %	Roll-off Angle as a Function of the Rate of Inclination of the Plane to the Horizon (°/s), °		
		0.37°/s	0.61°/s	1.1°/s
Poly-(GMA-co-TEMA)	29.0	10.8 ± 3.5	10.4 ± 3.2	8.7 ± 2.0
	44.4	9.7 ± 1.1	9.6 ± 2.3	8.9 ± 1.0
	62.1	5.6 ± 1.8	4.9 ± 2.2	3.5 ± 1.4
Poly-TEMA	100	4.8 ± 1.6	2.8 ± 1.6	2.6 ± 1.2
Poly-(GMA-co-HFMA)	29.0	15.7 ± 1.1	9.7 ± 2.8	8.6 ± 2.4
	41.7	7.8 ± 3.3	6.6 ± 2.6	5.7 ± 1.1
	61.7	5.4 ± 1.6	5.3 ± 1.6	4.8 ± 1.0
Poly-HFMA	100	4.7 ± 1.2	4.0 ± 2.0	3.5 ± 2.0
Poly-(GMA-co-HIMA)	29.8	8.8 ± 2.5	6.4 ± 2.5	5.8 ± 2.3
	42.7	7.8 ± 2.2	5.8 ± 2.0	5.0 ± 1.8
	64.9	5.8 ± 2.1	4.2 ± 2.0	3.4 ± 1.8
Poly-HIMA	100	3.7 ± 1.2	2.9 ± 1.5	2.3 ± 1.4
Poly-(GMA-co-HBMA)	29.3	6.1 ± 1.7	5.6 ± 2.5	4.5 ± 2.0
	46.7	5.3 ± 1.3	5.2 ± 0.6	4.5 ± 1.7
	65.8	5.4 ± 1.1	4.4 ± 1.6	3.7 ± 1.7
Poly-HBMA	100	4.1 ± 1.1	3.1 ± 1.3	2.8 ± 1.0

**Table 6 polymers-14-01960-t006:** Percentage of water droplets rolling off the aluminum surface modified with GMA- and FMA-based copolymers with a theoretical FMA content of 66.7% when the drops are dropped onto a plane with a fixed angle of inclination to the horizon.

Modifier	Percentage of Spontaneous rolling off of Water Droplets When They Are Dropped Onto a Plane with a Given Angle of Inclination to the Horizon, %		
	3°	5°	7°
Poly-(GMA-co-TEMA)	83%	93%	100%
Poly-(GMA-co-HFMA)	80%	91%	96%
Poly-(GMA-co-HIMA)	80%	93%	95%
Poly-(GMA-co-HBMA)	80%	93%	98%

## Data Availability

Not applicable.

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
