# Peer review of "Effect of the Composition of Copolymers Based on Glycidyl Methacrylate and Fluoroalkyl Methacrylates on the Free Energy and Lyophilic Properties of the Modified Surface"

_polymers, 2022, doi:10.3390/polym14101960_

Round 1

Reviewer 1 Report

This manuscript describes a study of using reactive copolymers based on glycidyl methacrylate and fluoroalkyl methacrylates as effective agents for reducing surface free energy (SFE) in producing self-cleaning polymer coatings on Al coupons. The study is well designed and presented. I have only the minor comments:

  1. Table 1 and 2: it is good to provide the equipment of analyzing the composition. I believe Table 1 is by CHNOS elemental analyzer while Table 2 is by EDS?
  2. Fig. 1: title of X-axis is better to change to “Molar amount of FMA(%)”
  3. Fig. 2: it is good to provide an image of bare Al substrate without coating. Fig. 2(b) seems not properly focused and is not sharp.
  4. Table 4: Error should have the same decimal place as the main data.
  5. Fig. 3(a &d): I would advise changing the decimal sign from “,” to “.”
  6. Fig. 7: It is good to provide similar set of Al coupon without such coating’s behavior after water droplets are rolled down from them. Currently all the soil particles just physically sit on the plate. Normal water flow should be able to wash some of them away. In reality, the dust particles are much smaller in size and may stick onto the plate for a long time under the curing of sunlight. It would be very excited to see the effect under such close scenario.
  7. The 1st sentence in Conclusion section consists of 5 lines and is super long. Pls rephrase it shorter.

Author Response

Thank you for attention and useful comments to our study.

We revised our paper. Changes in the manuscript are marked by yellow color.

«Fig. 2: it is good to provide an image of bare Al substrate without coating» - In our previously published works, there is the surface image of the original aluminum (for example, [47]).

«Fig. 7: It is good to provide similar set of Al coupon without such coating’s behavior after water droplets are rolled down from them. Currently all the soil particles just physically sit on the plate. Normal water flow should be able to wash some of them away. In reality, the dust particles are much smaller in size and may stick onto the plate for a long time under the curing of sunlight. It would be very excited to see the effect under such close scenario.» - In the case of smooth aluminum, the water droplets stick to the surface. The textured aluminum surface without a polymer coating exhibits superhydrophilic properties. 

«The 1st sentence in Conclusion section consists of 5 lines and is super long. Pls rephrase it shorter.» - We agree that the first sentence of the conclusions is long, but it briefly pepresents the idea of the whole work. So, in our opinion, the division into shorter sentences is inapplicable.

Reviewer 2 Report

This work reports on the utilization of reactive copolymers based on fluoroalkyl methacrylates and glycidyl methacrylate to reduce surface free energy (SFE). The manuscript is part of a series of studies concentrated on the manipulation of lyophilic characteristics of the surface, providing insights into the effect of the structure and composition of copolymers on the SFE and water-repellent properties of coatings. The optimized copolymer compositions have the potential to be used in the fabrication of self-cleaning polymer coatings. The study is well designed and nicely presented. I have only few minor comments.

  1. Page 16, line 470. I did not see “a-f” in Fig 7. I believe it is “Figure 7 (1)-(7)” ?
  2. It is good to provide the time of every single image in Fig 8 in the caption, though the movement of a water droplet is well presented in the video.

Author Response

Thank you for attention and useful comments to our study.

We revised our paper. Changes in the manuscript are marked by yellow color.

Reviewer 3 Report

The investigation performed by Klimov et al. is a great piece of work. The authors have fabricated self-clean coating materials based on a number of copolymers of glycidyl methacrylate and fluoroalkyl methacrylate with varying fluorine atoms. The authors have discussed the outcomes very nicely and the videos are very attractive. However, a few points are need to be addressed before the publication of this manuscript. I strongly recommend the publication of this manuscript after the minor revision.

  1. The introduction section seems to be weak. The authors have discussed only about Cassie and Wenzel states. A few recent literatures investigating a similar work or on self-clean coatings need to be discussed. The below pertinent references can be considered:
  2. https://doi.org/10.1080/03602559.2018.1447128
  3. https://doi.org/10.1016/j.molstruc.2019.127342
  4. DOI: 10.1039/B412803F
  5. https://doi.org/10.1016/j.porgcoat.2019.105306
  6. https://doi.org/10.1021/acsnano.5b04230
  7. https://doi.org/10.1016/B978-0-12-816671-0.00001-1
  8. https://doi.org/10.1016/j.nanoen.2016.11.020
  9. https://doi.org/10.1007/s11831-021-09689-1
  10. All the parts in Fig. 1 can be clubbed together with single x- and y-axis legends (Work of adhesion and amount of FMA%). The authors don’t need to put these legends for each part as all of them are the same. The same can be done for the remaining figures wherever required.
  11. In Fig. 5 legend, the authors have mentioned CA, though it is well understood that it represents the contact angle but I request the authors must define the acronym at least once in the manuscript (preferably at its first use).
  12. By any chance is there any possibility to show the tilt angle in Figure 7?

Author Response

(The authors gave the same response as above.)
